# Detecting Physical Impacts to the Corners of Shipping Containers during Handling Operations Performed by Quay Cranes

**Sergej Jakovlev** [1,2,*], **Tomas Eglynas** [1], **Mindaugas Jusis** [1], **Miroslav Voznak** [1,2], **Pavol Partila** [1,2] **and Jaromir Tovarek** [1,2]

1   Marine Research Institute, Klaipeda University, Herkaus Manto Str. 84, LT-92294 Klaipeda, Lithuania
2   Department of Telecommunications, VSB—Technical University of Ostrava, 17, Listopadu 2172/15, Poruba, 708 00 Ostrava, Czech Republic
*   Correspondence: sergej.jakovlev@ku.lt

**Abstract:** This study aims to address the problem of proper shipping container damage detection during the hooking procedure performed by quay cranes and their hooking mechanisms. We adopted the Impacts Detection Methodology (IDM), developed previously, to detect repeated impacts on the same areas of the container. These concurrent impacts indicate false hooking procedures, which result in dangerous metal deformations in these areas over short periods of operational time. This application intends to verify if this methodology is adaptable in real-life applications to detect these specific events. Our main results indicate that more than half of handling procedures are carried out with a higher risk of structural damage to the containers due to these repeated impacts, which can reach up to five concurrent impacts in some case studies.

**Keywords:** acceleration; sensor; shipping container; security; physical impact

## 1. Introduction

### 1.1. Problem Statement

Container handling and transportation, particularly during trans-shipments, is a challenging technological challenge that has not been fully solved up to this point [1]. The majority of the containers being transported today are out-of-date, and even the newer containers are being developed following older outdated standards, making them unsuitable for smart autonomous and highly secured operations. This presents a significant challenge for the companies and the engineers working on the repairs [2]. The spreader's faster velocity and high force impacts on the corners of the containers when they are hooked by quay cranes (QC) from AGVs or the inside of the container ships; these are the most important physical effects that happen during container trans-shipment in port operations [3]. The results of continuous impacts near the hooking mechanism from the outside are shown in Figure 1a below, while the results of continuous impacts by the spreader from the interior of the container are shown in Figure 1b. Given that QCs are run by human operators, mistakes will inevitably occur during the workflow. The hooking procedure has a technical problem. Due to the operator's lack of the aforementioned experience and the unique and unpredictable environmental conditions that exist just before the final hooking stage, it is impossible to precisely direct the spreader hooking rods to the container that is in the ship's hull (or underneath the crane) to hook it.

Consequently, certain areas of the container are prone to damage due to impacts. Figure 2 below shows the operational environment of the hooking technique and the container's movement direction. Due to various circumstances, the spreader hooking mechanism is not dynamic, which makes it impossible to attach it simultaneously to all four fixation points.

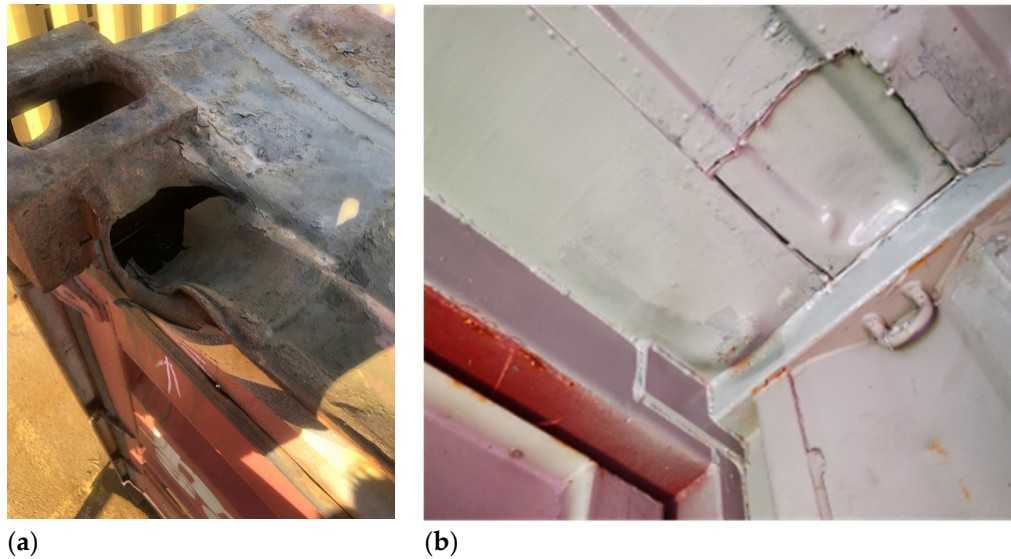

(**a**)                                    (**b**)

**Figure 1.** Results of the spreader hooking mechanism's impact on the container's incorrect hooking locations. Here figure (**a**) shows the result of the continuous impacts on the exterior part of the container, and (**b**) shows the result of these impacts on the interior part of the container.

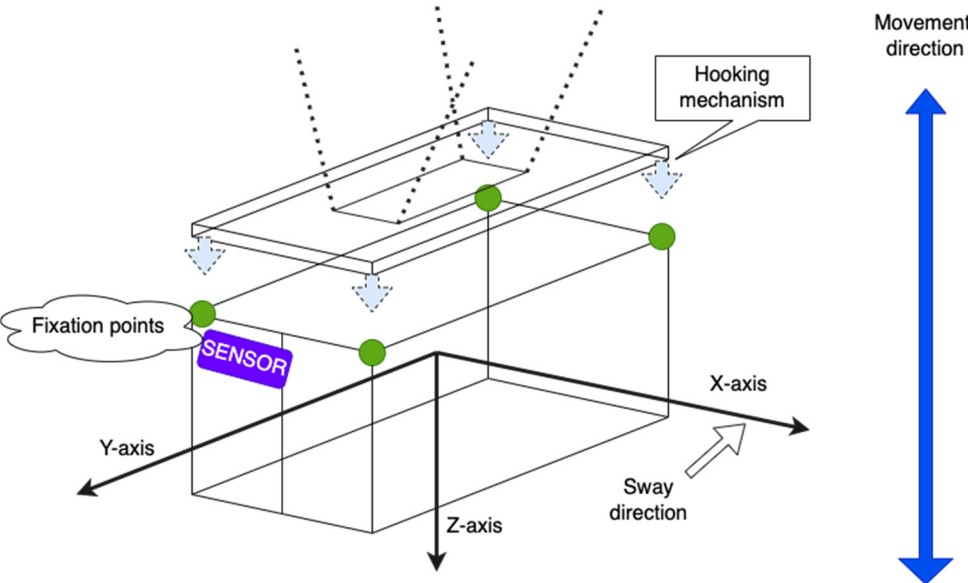

**Figure 2.** Diagram showing the path taken by the container during experimental case studies.

Figure 3 below schematically shows the points of impact and their cause. The container swings unpredictably, causing deviation in the X-axis movement. This results in the container being struck by the hooking mechanism close to the fixation point, causing severe damage to the metal structure already shown in Figure 1. This happens due to various factors, such as weather conditions, human factors, dynamic properties of the control mechanisms, dynamics of the crane structure, and the lines holding the container, among others. These impacts occur randomly and cannot be predicted, nor can they be avoided by using the same mechanics, control strategies, and operators for the handling operations.

Due to harsh working conditions, high optimizing costs of the heavy machinery, and not yet well-developed control technologies, these impacts will continue and will cause many more losses in the long-term perspective due to the increasing number of container trans-shipments in the world.

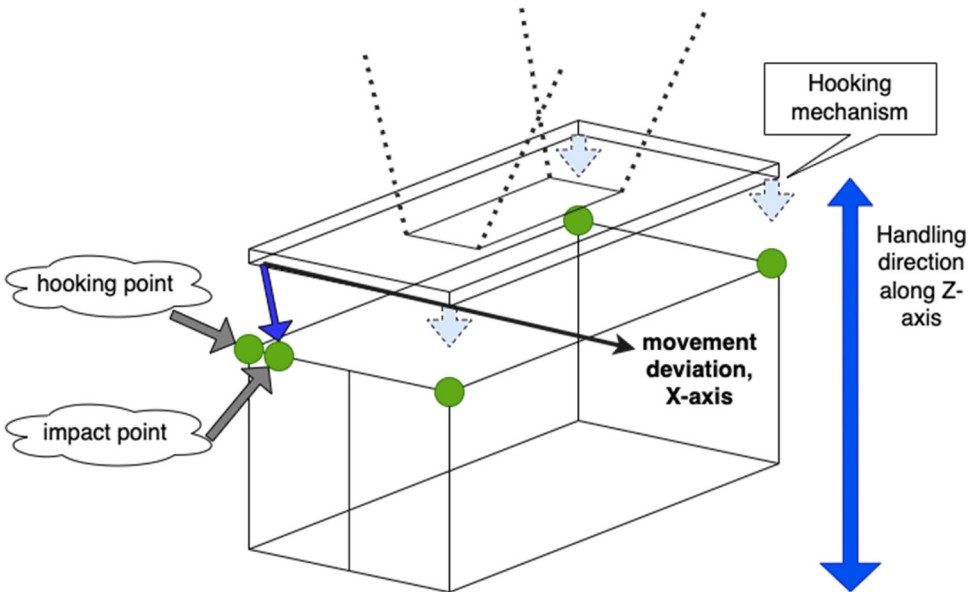

**Figure 3.** Schematics of the impacts' causes.

### 1.2. Analysis of Advancements

In close collaboration with Klaipeda container terminal representatives, a preliminary investigation revealed that these consequences are the most significant and persistent, leading to longer transit delays, higher repair costs, and a larger risk of subsequent damage to surrounding workers and containers [4,5]. This paper carries on the team's research efforts in the area of physical impact detection (acceleration signal extraction, pre-processing, smart analytics, and transfer as result) at specific times and locations of contact, during container handling operations carried out by the spreader of the quay and yard cranes. The new application is ensured to have the highest level of adaptability, accuracy, and stability. Jakovlev et al. [6] introduced a novel methodology for impact detection during inner hull operations with loaded containers. The goal of this paper is to further develop the research foundation and widen the scope of detection, specifically regarding the hooking accidents that pose serious safety concerns for shipping companies, as well as to ensure that the issues they address are handled appropriately in line with applicable safety regulations and as effectively as is technologically possible in the highly demanding work environment. Based on the extensive list of detection results, the goal of this paper was realized.

To increase the security of the logistics, the safety of the technical processes, and the safety of the cargo, new, remote, and time-effective container inspection techniques are essential. Theoretically, similar problems may be addressed from a methodological perspective if the emphasis is placed on the measurement of vibration signals rather than on pinpointing the location of the impact. Such cases include ultrasound inspection, smart visual inspection to test for container damage, and the search for the resonance frequency of a material in the case of continuous pulsations induced on the metal structure of the container walls that could result in its collapse if certain threshold values are met. Capacitive sensors attached to self-oscillating circuits, for instance, could be used to detect metal deformations in structures under different stresses or by more in-depth analysis of the structure's initial vibration [7]. Another research trend is the adoption of expensive developed solutions; Chuan et al. [8] were able to measure with precision the delay in arrival times between two connected sensors and the propagation of impact noise through the material by adjusting to the model of vibration of a tight membrane [9]. Yet, auto-detecting of potential damages in the heavy transportation industry has been prioritized before by several authors, most notably by Molodova et al. [10], who presented an automatic method for detecting railway surface defects, called "squats", using axle box acceleration (ABA) measurements on trains, and Wiseman [11] suggested a safety tool for an incessant inspection of "SkyTran" tracks by

employing digital surveillance technology. The damage detection is done manually by visual inspection on-site, after a critical event, or just as part of a regular check [12]. If the inspection of the ship is done regularly, the risk of vertical cell guides damage caused by containers in the later transportation process can be avoided. Yet, this detection method is not only time-consuming and labor-intensive but also places high demands on specialized personnel working aboard the ships [13]. Many researchers are studying methods and developing complex sensory systems [14] to assist or replace manual infrastructure inspection aboard ships, including optical devices, laser scanners, etc. Applicable digital signal processing algorithms are a hotspot of data science research [15,16]. Here, Gabriel et al. [17] focused on the detection of impacts and flaws of structures in airborne vehicles with an introduced electronic Structural Health Monitoring (SHM) prototype in which the analytical part was based on the propagation of ultrasound acoustic waves. Other researchers such as Andrzej and Sandris [18] explored the effectiveness of damage identification in composite plates using damage indices based on smoothing polynomials and curvelet transform, and Qian et al. [19] investigated the technical and computational possibilities for early fault detection of rotor unbalance yielding incipient rub-impact faults in the initial stage based on the deterministic learning method. Finally, Natalino Daniele et al. [20] demonstrated state-of-the-art impact detection techniques for aerospace structural components, combining the most promising techniques currently available: the first one was related to impacts inducing low-frequency vibrations based on Neural networks (NN), while the second one was related to the detection of impacts inducing higher-frequency stress waves based on an acoustic source localization approach. In all related case studies, the detection of metal structure deformation is mostly done by employing visual inspection, if there exists the possibility to inspect the area of impact closely, to be sure that a certain event happened [12]. However, this is still an emerging field. Applicable in a real operational environment, digital signal processing systems and trustworthy algorithms are a hotspot of data science research [15,16] and will remain so in the coming decades and will closely correlate with the means of delivering safety and security to container handling operations and to trans-shipments [21].

Regarding the safety and security of the containers and the cargo, a few research efforts have been reported in the literature [21]. However, none of the earlier publications cover the use of acceleration parameters to find impacts while handling actions are being performed. The materials, type, and shape of the infrastructure, the speed of the handling procedures, the mass of the container, the physical characteristics of the suspension system, as well as the inclusion of the human factor, and the general dimensionality of the test-site, all have an impact on the patterns of container sway and natural vibrations of metal structures.

The research group has partially dealt with similar matters in several research projects recently, developing intelligent container system components and parametrizing the efficiency, safety, and security of handling procedures of the port operations, resulting in several high-impact publications [6,22–24]; yet, detection of the impacts to the exact positions of the container structure using vibration criteria or any other evaluation criteria has not been analyzed before, either by the authors or by any other research group. It is a novel topic for the engineering community as well as a huge opportunity for companies to start solving these critical problems. Therefore, this paper aims to develop a novel unified impact detection system and method, and this paper hereafter will summarize the research findings of our group using the data sets collected using the technological equipment from our previous studies, while detecting the abovementioned impacts.

### 1.3. Definition of the Dynamics

Figure 4 depicts one of the registered crane spreader paths in one hooking (handling) cycle. The data were obtained using a digital accelerator recording device attached to the spreader, based on which the acceleration values and the path of the spreader were calculated. The spreader path consists of several typical stages:

- Starting with point one, the spreader is unhooked from the previously transported container;
- Then going to point six, the vertical lowering of the spreader towards the container begins;
- Ending with point seven, the spreader finally touches the container, which is followed by the physical hooking to the container using the contact points.

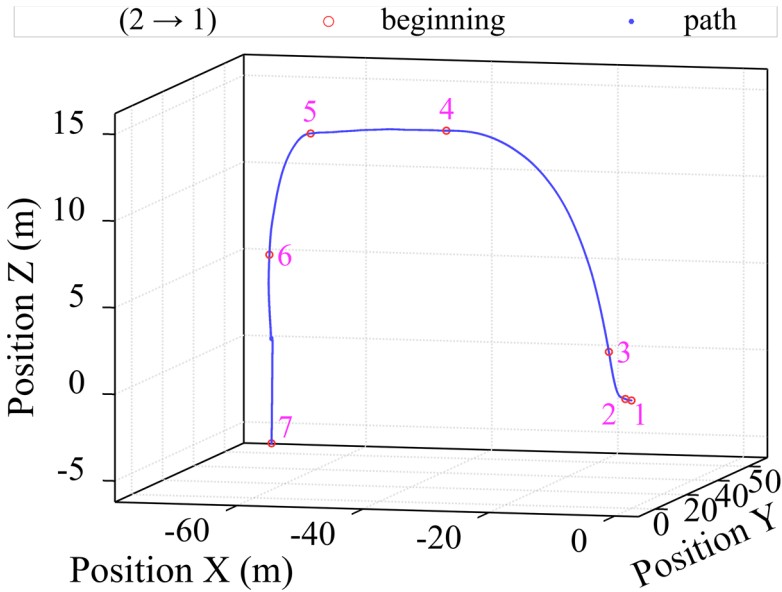

**Figure 4.** Container crane spreader trajectory in one handling cycle.

In this study, we examined only the processes occurring after the sixth stage of the full cycle.

Figure 5 depicts one of the many vertical positions of the spreader calculated based on the recorded accelerations over time. It can be assumed that in the 3rd second, the operator slows down the lowering process of the spreader targeting the exact ship's container shaft. From the 16th second, the spreader is again lowered down with increased speed until it reaches the container at the start of the 22nd second.

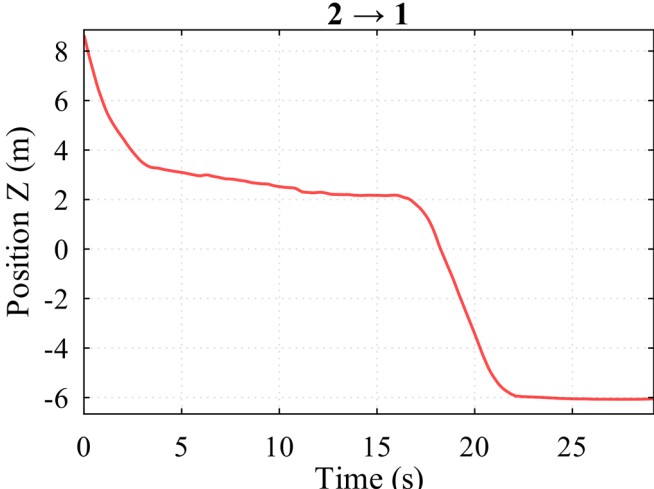

**Figure 5.** The way of the spreader in the dimension of the Z-axis: Example A.

A more detailed course of events is revealed by the recorded accelerations of the spreader along the vertical Z-axis, which are presented in Figure 6. Looking at the graph, we can assume that the sudden increase in acceleration at the start of the 12th second marks the beginning of the spreader trajectory within a container shaft of the ship. The

sudden change at the 23rd second is very likely to mark the initial physical contact with the container, and the following small deviations are a consequence of the work of the hooking mechanisms. In this example, the operator successfully hooked the container on the first try.

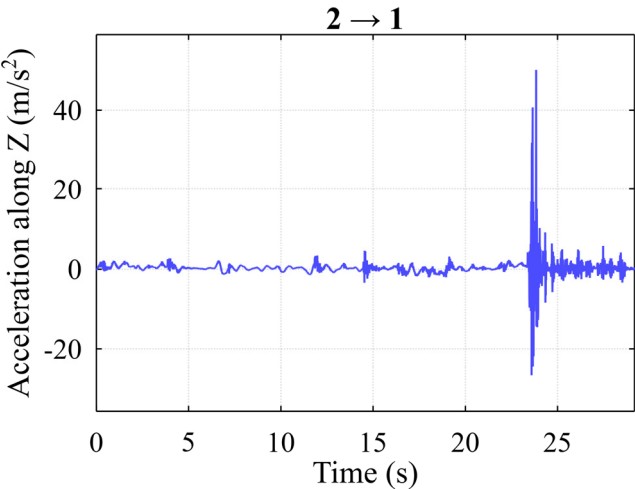

**Figure 6.** Spreader accelerations along the A-axis: Example A.

However, successfully hooking sequences are not casual. This is illustrated in Figure 7, showing the path of the spreader. It is visible that there are repeated attempts to hook the container in the period between 3.5 and 5.5 s.

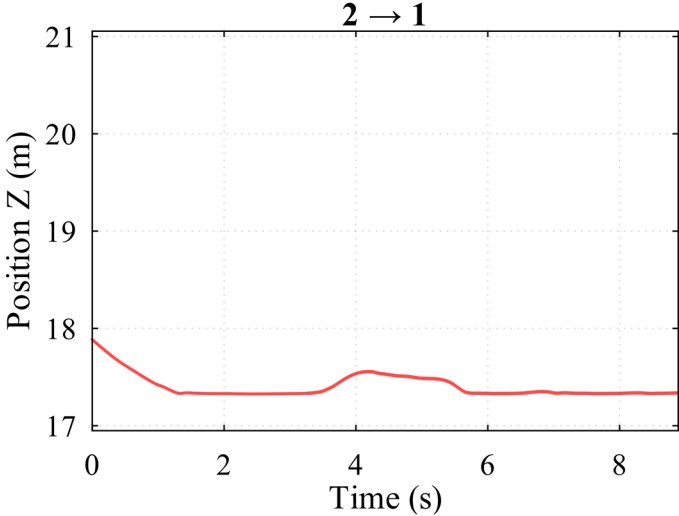

**Figure 7.** Spreader trajectory along the Z-axis: Example B.

Analysis of the acceleration values in Figure 8 gives more detailed results. At the start of the 1st second, the initial impact occurs when the spreader impacts the container with acceleration, which in turn is followed by other attempts to hook the container—with other impacts. This is observed from the start of the 3.5th second, where unsuccessful spreader-lowering processes occur, ending with a secondary impact at the start of the 5th second.

To carry out serious container damage assessments, it is necessary to trace the impact events, either small or big, as demonstrated earlier. Companies having real-time event-based information would be able to immediately repair minor damages to the hooking mechanisms of the containers, prolonging their usage without halting the transportation cycle of each separate cargo.

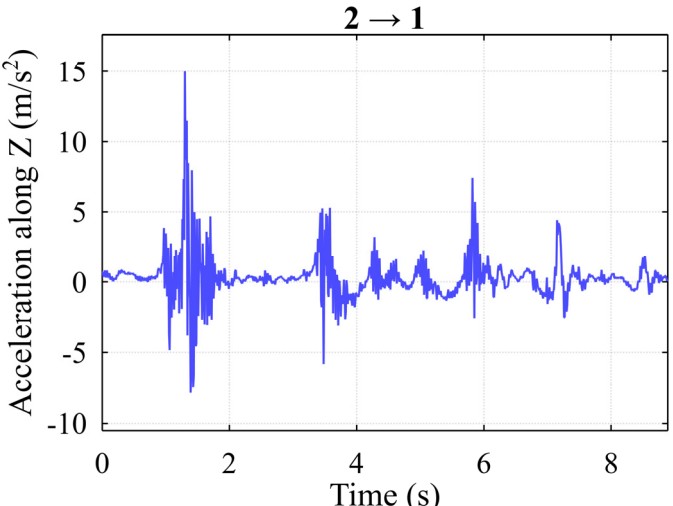

**Figure 8.** Spreader accelerations along the Z-axis: Example B.

## 2. Methodology

### 2.1. Impact Detection Methodology: Introduction

For the registration of repeated impacts of the spreader into the containers, a method was adopted, namely, the Impacts Detection Methodology (IDM), which is based on the analysis of the vertical acceleration parameter. In general terms, the IDM signal processing algorithm embedded in the system is presented in Figure 9. This method was initially developed by the authors using historical data for the detection of the true thresholds and was discussed in detail in [6]. The proposed method of detecting crane spreader impacts into a container consists of several stages: filtering the signal, finding peak values in the filtered signal using statistical analysis, selecting the threshold steps, distinguishing potential moments of impacts (events) using the threshold value, excluding the actual critical impact events by grouping all the potential events, and determining repeated impacts in the same area.

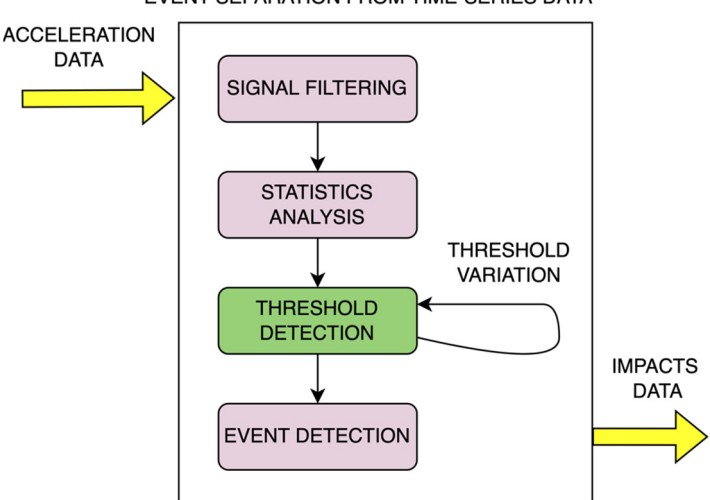

**Figure 9.** Flowchart of the proposed solution for event data detection.

The IDM method uses a threshold detection algorithm, which detects the desired optimal threshold level to be used as a starting point for impact detection, the event frequency, which corresponds to 100% at this point. While the algorithm is based on the method provided earlier, the IoT system differs, with the detection unit mounted on the corner of the container, nearer to the impact point. The following Figure 10 demonstrates

the used IoT framework with the hardware module, including the digital accelerometer, as well as the main evaluation criteria in the experimental environment. While other computational and electronic elements are hidden in this research, they are discussed in the previous work by the authors.

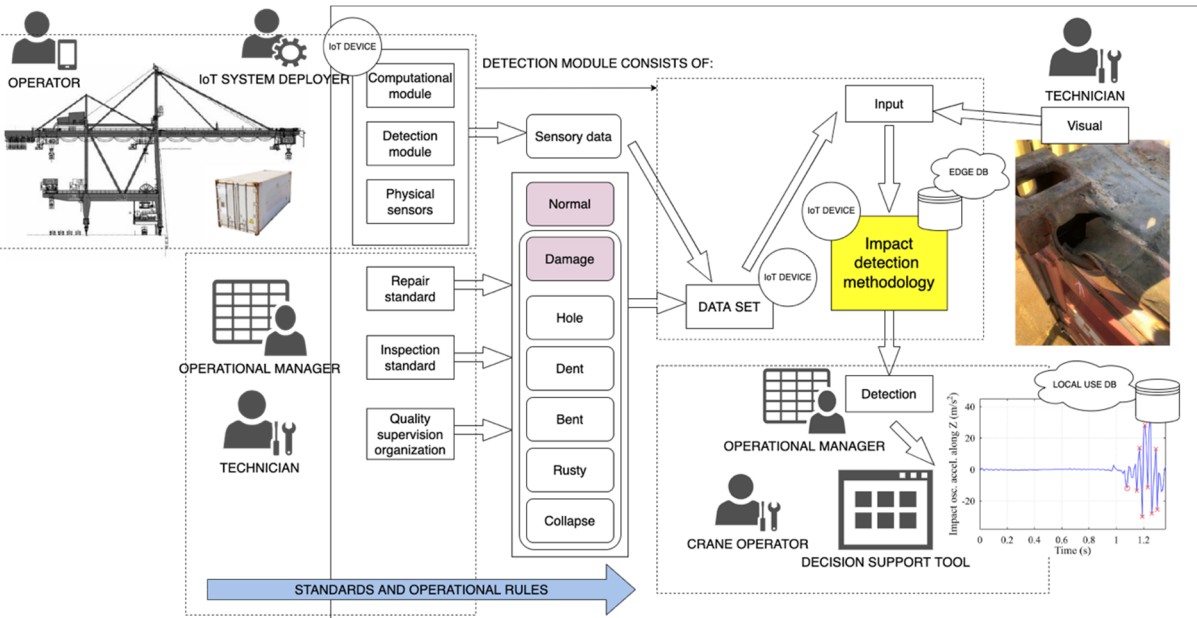

**Figure 10.** Impacts detection framework used in the study.

Verification model logic includes the IDM math logic with varying acceptable acceleration parameter threshold values. Prediction accuracy in this case depends on the acceptable filtering technique. The categories for each impact can be classified into several important events: bent, dent, or hole in the structure. Of course, an impact ordinary can generate little to no none harm to the metal surface; therefore, impact strength and velocity are crucial factors in this kind of estimation in real-time.

When the crane transports its spreader, it vibrates at its characteristic natural frequency. Since the vibration amplitude is small compared to the length of the lines holding the container, the curve of the vibration in the plane is close to the sinusoid. When assessing the length of the spreader lines, the frequency of vibration is low. However, during handling phases, the spreader is affected by external forces, for example, transitional processes of winding of the spreader lines or contact of lines with external structures, which complement the range of movement of the spreader with components of different frequencies.

The speeds and accelerations of the spreader's transportation process are limited, so the frequency of the spreader's swing should be low. In the generalizing spectrum of vertical acceleration (Figure 11), compiled by analyzing the 102 cycles in the period of interest, the components of the low-frequency acceleration signal dominate. It can be said that the spectrum up to 5 Hz is flat enough.

We are interested in events when the spreader comes into contact with a container or other external structures. During contact, the direction of movement of the spreader changes, which is also reflected in a sharp change in the value of its acceleration; for example, in Figure 12 (3rd fragment at the 23rd second of Figure 6), at timestamps 1 s, 1.1 s, etc., these changes in the acceleration spectrum "generate" higher frequency components.

A glance at the spectrogram (Figure 13, where the darker the red color, the stronger the signal) of the acceleration shown in graph 6 confirms that at the 23rd second, there was an impact on the container: the acceleration signal dominates the spectrum section. On this basis, it would be possible to determine whether there was contact between the spreader with the container.

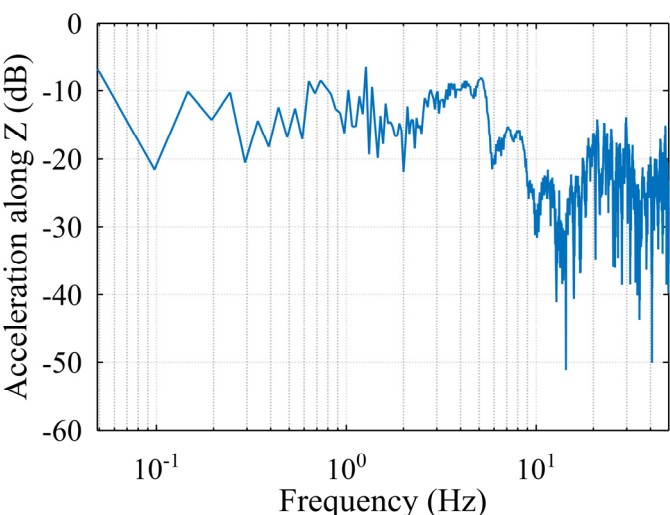

**Figure 11.** Acceleration spectrum along the Z-axis.

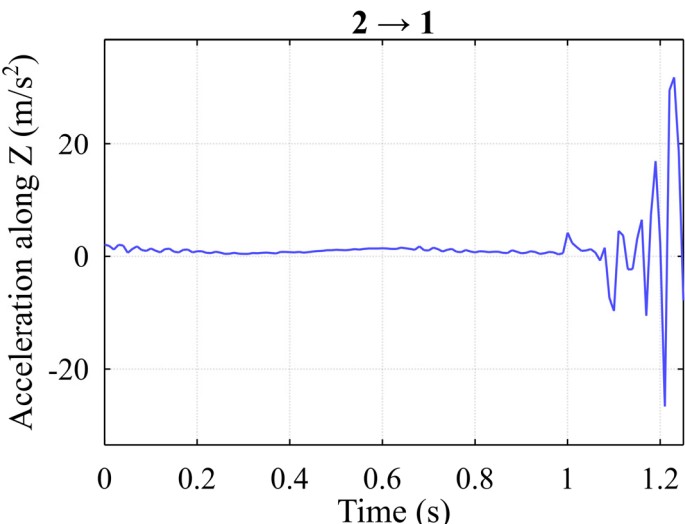

**Figure 12.** Shock moments along the Z-axis during movement in the hull of the ship.

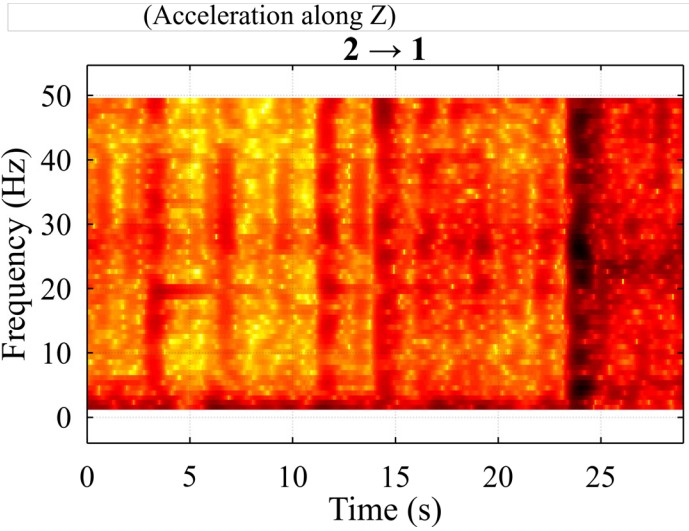

**Figure 13.** Acceleration spectrogram along the Z-axis.

### 2.2. Detection of Impact Events: Background

As was already noted, impacts create higher-frequency components in the acceleration signal; hence, these oscillations are isolated for further study with the aid of a high-frequency filter (high-pass filter). Here, a filter that has the lowest possible phase distortions, a constant delay over the frequency range, and a monotonic amplitude response across the range of passing frequencies is required to reduce the distortion of the signal for further analysis (1).

$$H(z) = \frac{B(z)}{A(z)} = b_0 + b_1 z^{-1} + b_2 z^{-2} + \cdots + b_N z^{-N}. \tag{1}$$

where *N*—the filter row and *b*—the coefficients that are obtained using the GNU Octave function "fir1".

A step-by-step methodology was created that first uses the zero-crossing approach to discriminate between positive and negative half-waves to determine the amplitudes of the acceleration oscillations' half-period oscillations. Positive and negative half-wave data arrays were combined into a single one. With this information, the half-wave extremums were sought:

$$[p, p_{\text{idx}}] = \mathop{}_{i\,=\,1}^{\text{M}} |max(|a_f(S_i : E_i)|), \tag{2}$$

here:

- $a_f$—an array of filtered accelerations;
- $S_i$—an array of the half-wave indices;
- E—an array of the half-waves end indices;
- M—the number of the half-waves;
- $p$—the values of the extremums found;
- $p_{\text{idx}}$—the positions of the found extremums in the array of the filtered accelerations.

A fragment of the results is presented further.

Figure 14 shows the signal oscillating around the zero value (axis), which was obtained from the vertical acceleration signal recorded by the sensor, after the application of the high-pass filter. The figure also shows the peaks of the half-periods of the signal oscillations marked with green circles, which were found using the step-by-step methodology.

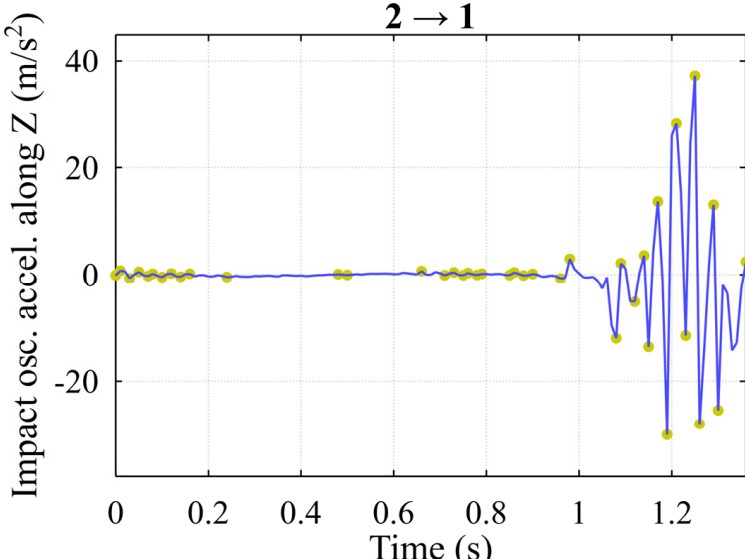

**Figure 14.** Peaks of acceleration oscillations.

Each peak discovered in the earlier stage may be a result of the collision. The filtered signal's threshold value was utilized to determine whether peaks corresponded to the crucial impact; peaks with absolute values over the threshold indicate the potential moments

of the events. Based on data distribution function criteria that best reflect the peak values, statistical analysis was utilized to establish the threshold value. The most frequent peak values, which can be termed noise, are in the region of their maximum and are shown in Figure 15 as green circles. The threshold relative value of the event's frequency, which at the point equates to 100%, is a key evaluation criterion, according to IDM.

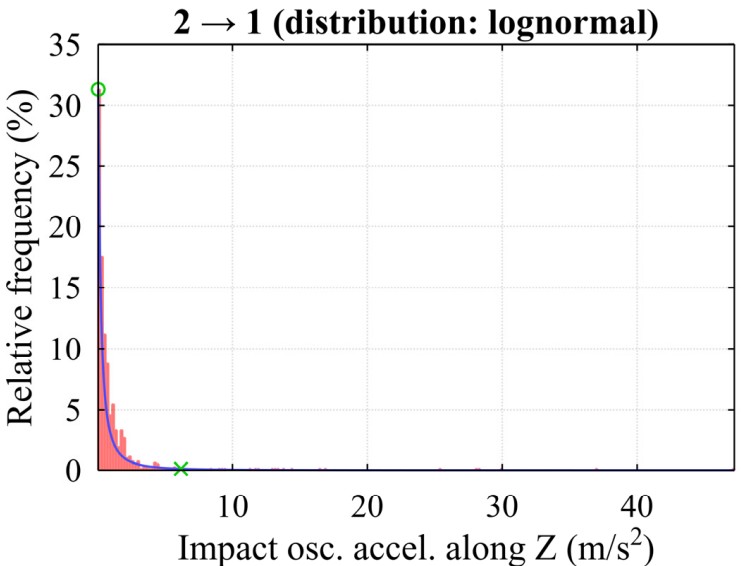

**Figure 15.** Histogram of signal peaks.

When the value of the parameter drops as peak amplitudes increase, the signal experiences fewer of these peaks. We obtained an acceleration threshold value following the function of the distribution after picking a specific value for the parameter (red cross in Figure 16).

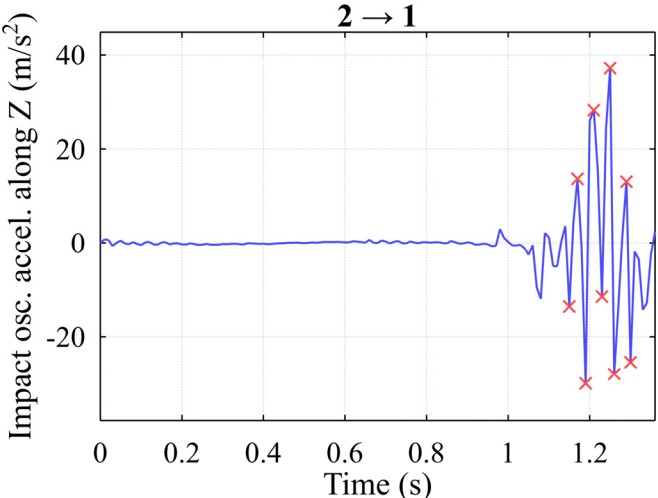

**Figure 16.** Detected potential impact events (here red crosses indicate the peak values).

Peaks that have amplitudes greater than the threshold were chosen from the sample of detected peaks using the resulting acceleration threshold value, giving us peaks that indicate potential occurrences (in Figure 11). Potential impact events were grouped using the established methodology, and as a consequence, distinct impact events and their starting points are obtained (marked by red circles in Figure 17).

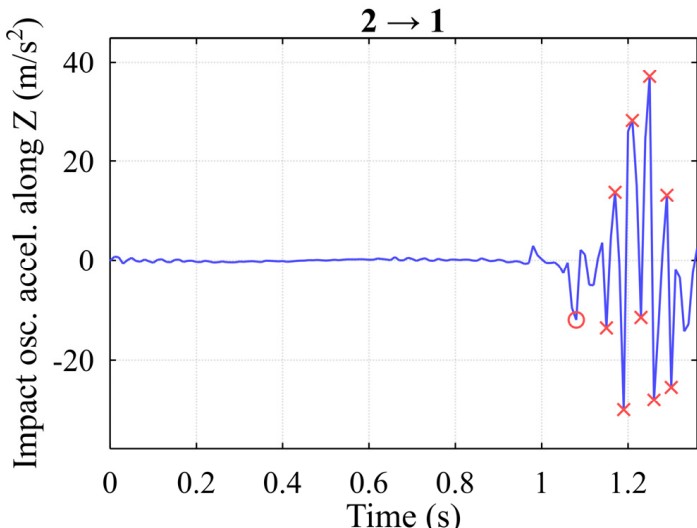

**Figure 17.** Example of an extracted initial impact event (here the red circle indicates the starting point, and the circles indicate the resulting peak values of the potential impacts).

Figure 18 shows the original acceleration curve (from Figure 6) with projections of the events presented above.

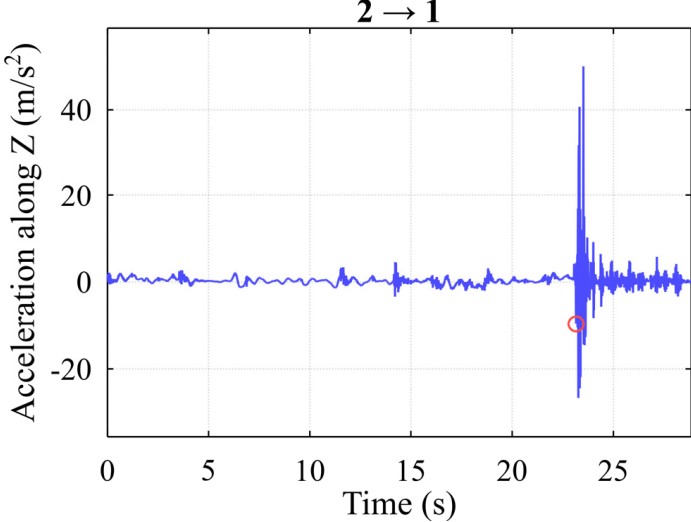

**Figure 18.** The event at the main signal (here the red circle indicates the starting point of the initial impact).

### 2.3. Acquisition of Optimal Parameters for Impacts Detection

The filter corner frequency and the threshold relative value of the event frequency are the two selectable parameters used by the approach, as was previously indicated. The results produced by the approach can be altered within broad bounds, and the results also vary within broad bounds: for instance, by choosing which values, a sizable number of registered events can be acquired (Figure 19). The research was done, during which the parameter values were adjusted within broad boundaries, and the results produced by the method were assessed, to identify what values of the parameters influence the results acquired by the method that best match the impact events.

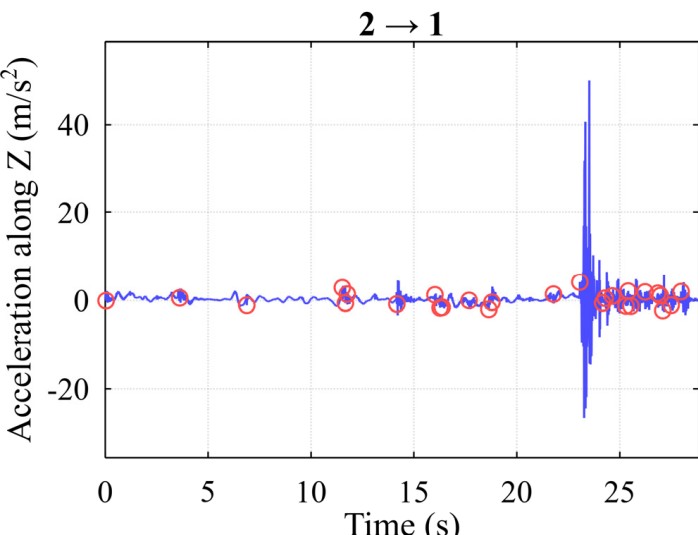

**Figure 19.** The result is from an incorrect selection of parameters (here the red circles indicate the registered events).

This method was used in this study to evaluate 102 occurrences of intermodal crane hooking and handling. The sequence of values collected was summarized by computing the RMS of the results. Figure 20 shows how, as the filter frequency changes, the *p*-value of the chosen distribution function (obtained by the Chi-square goodness-of-fit test) changes. The RMS value remains above 0.15 in the low-frequency region up to 4 Hz. According to the spectral acceleration analysis (Figure 11), the high-amplitude acceleration oscillations we are interested in are at lower frequencies, so the corner frequency of the filter should be determined based on the local maximum at lower frequencies.

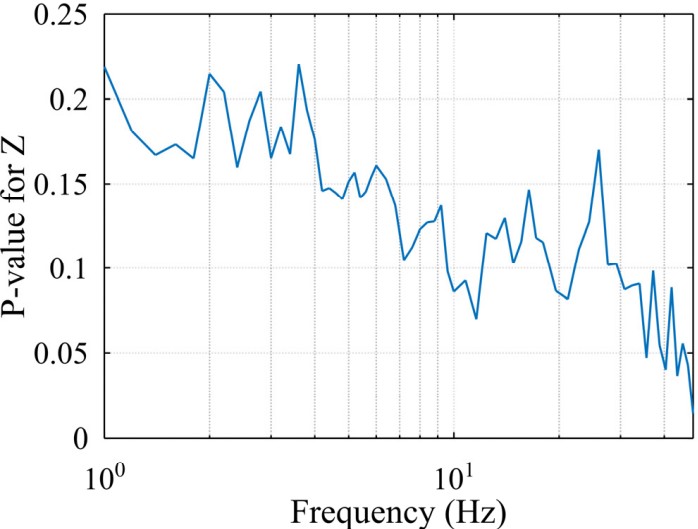

**Figure 20.** Variation in the *p*-value along the frequency range.

Figure 21 shows the number of possible observed events along the Z-axis of movement. As the filter frequency is increased, the period of acceleration oscillations is shortened, leading to an increase in the number of identified peaks for the same period. Theoretically, the important container-damaging vibrations should produce low-frequency, high-amplitude acceleration oscillations, as revealed by the spectrum analysis (Figure 11). This size alone is not appropriate for parameter selection, though, since the graph suggests that larger frequencies should be used.

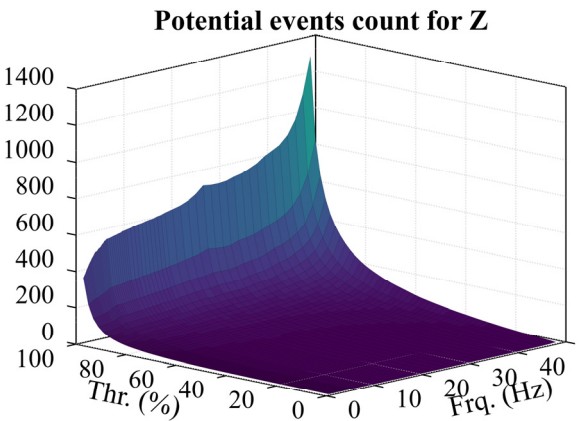 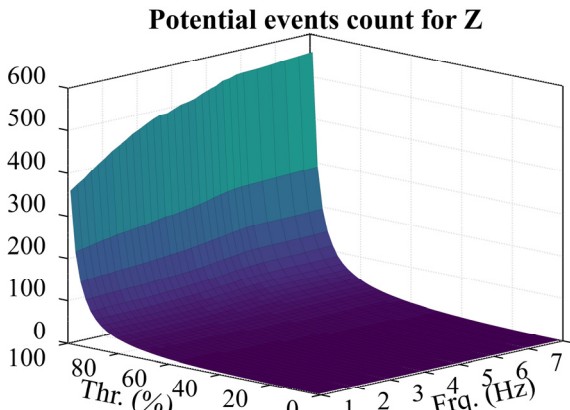

**Figure 21.** Quantities of potential impact events for different frequency ranges along the Z-axis.

It is evident from Figure 22 that as the filter frequency is increased, more possible event groups are detected along the Y-axis of movement parallel to the crane. This is because when the frequency rises, the amplitude of the filtered acceleration oscillations drops, perhaps increasing oscillation bursts and so posing a risk of more clusters forming. Additionally, the graph indicates a pretty constant number of registered events in the range of 1.2 Hz to 2 Hz at the beginning of the filter frequency range (looking at the extremes), and the amounts only rise as the frequency increases. This enables us to assume that the frequency response of the filter should begin at the value stated in subsequent research. The prior incorrect conclusion—that the frequency of the filter and the percentage threshold should be increased—is verified when looking at the number of identified events. However, it is doubtful that lowering the grab will result in more than 10 significant impacts to the container, hence it is not worthwhile to use filter frequency values.

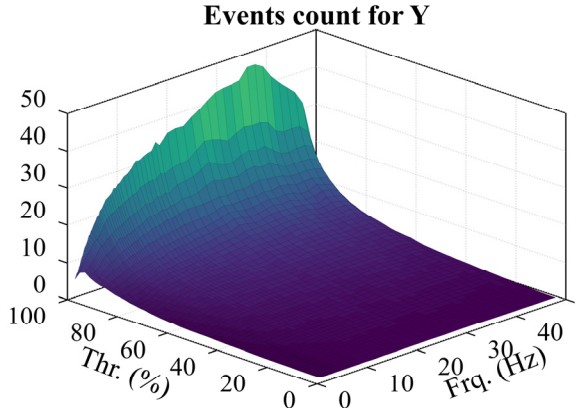 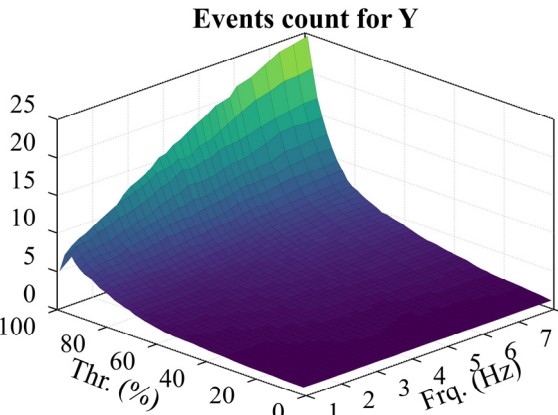

**Figure 22.** Analyzing the overall number of impact events concerning the intended threshold levels and frequency ranges along the Z-axis.

Additionally, it can be inferred from the acceleration spectrum (Figure 11) that high amplitude oscillations occur in the frequency range of up to 4 Hz (refer to Figure 23). This frequency range will be used in the search for the best results. A weighted amount, or the sum of the number of events and the RMS average of probable occurrences, was suggested as the ideal parameter value in a prior study of container impacts on vertical cell guides, and it produced good results.

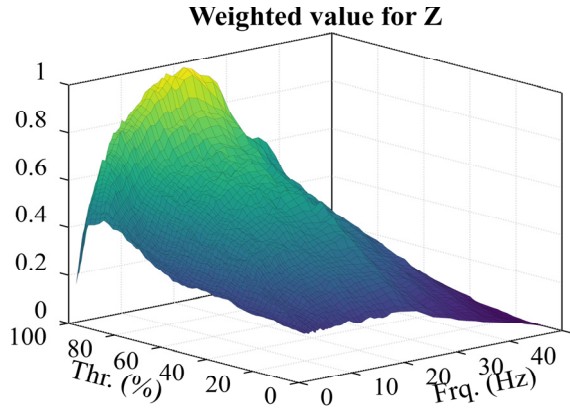
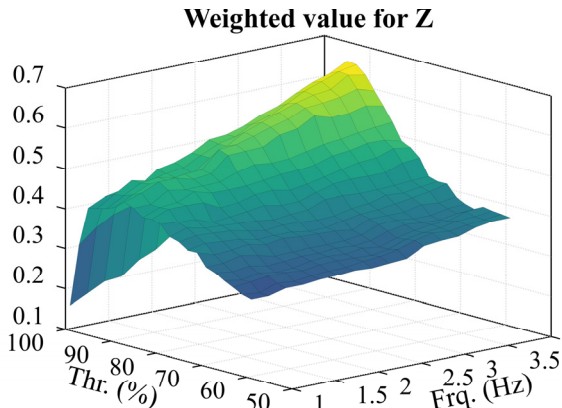

**Figure 23.** Estimates of weighted values concerning the variable frequency range (on the right, a narrower section of frequencies and relative threshold parameters is shown in comparison with the one on the left) along the Z-axis.

Therefore, going ahead, this study will also employ the aforementioned estimate. It can be seen that the weighted maxima vary little, except for the filter frequency range of 1.2 Hz to 2 Hz, and that a local extremum is present here at 1.6 Hz and 8%.

### 2.4. Detection of Repeated Impacts (Re-Impacts) to the Same Areas

The main concern of the research is the detection of repeated impacts during a single handling procedure. After examining typical spreader descents with at least one re-impact with the container (e.g., Figures 7 and 8), the following assumptions were made for re-impact detection:

- The vertical acceleration oscillations of the primary impact have the largest amplitude;
- The amplitudes of the re-impact oscillations are smaller, yet close in comparison;
- Each next impact stays almost identical in oscillation to the second one.

In the first step, the repeated Impacts Detection Methodology collects the maximum amplitudes of filtered acceleration oscillations of all registered impacts, i.e., the maximum amplitude of the filtered accelerations is sought in the period of each event. In the second step, the selection is made for the events with the maximum amplitude of the acceleration oscillations. This impact event will be considered the primary impact. In the third step, potential secondary impacts are selected, whose maximum acceleration oscillation amplitudes exceed the set threshold limit. Only those events that follow the primary impacts are selected from the number of potential detected secondary impacts.

The desired threshold level is a relative parameter calculated according to the maximum amplitude of the acceleration oscillation of the primary impact event. Statistical analysis was used to determine the limit of the threshold and the following assumptions were made. It is assumed at first that repeated impacts (re-impacts) are very rare (the cargo company's policy is to minimize the risk of damage to containers as much as possible by raising the qualifications of operators and establishing safe loading conditions); therefore, the upper limit is calculated based on the values of the amplitudes of the maximum acceleration oscillations of all events, and a relative threshold value is calculated from this detected value.

Going in the direction of the higher peak amplitudes, as the value of the parameter decreases, such peaks occur less frequently in the signal. The proposed method declares that after selecting a certain value of the mentioned parameter marked with a green cross, we obtain the acceleration threshold value according to the lognormal distribution function. Figure 24 demonstrates the initial close examination of all 102 handling cycles with the threshold level reaching 39.1% (green cross) according to the histogram of mean values distribution.

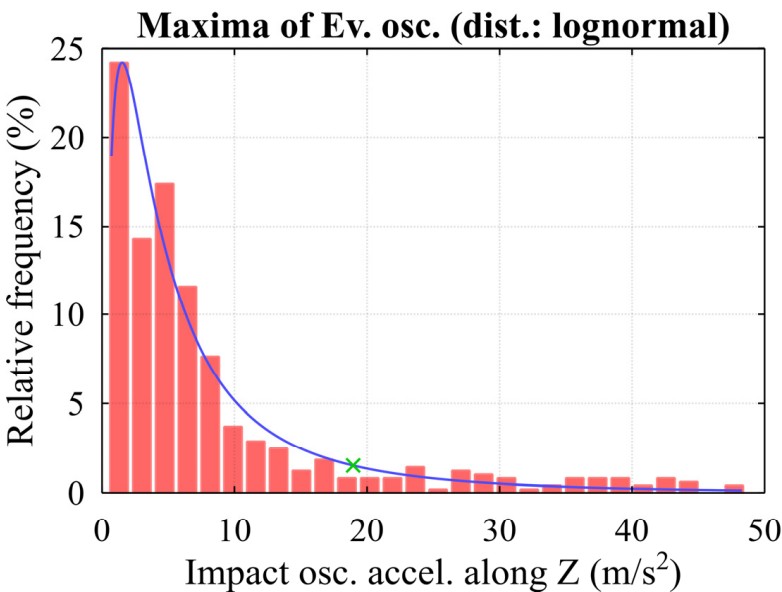

**Figure 24.** Histogram of the maximum amplitudes of the events.

## 3. Results and Discussion

The following Figure 25 demonstrates the results of the adoption of the proposed methodology with the selected optimum values from the previous section and is shown in Figure 8, where the secondary impacts are marked with squares.

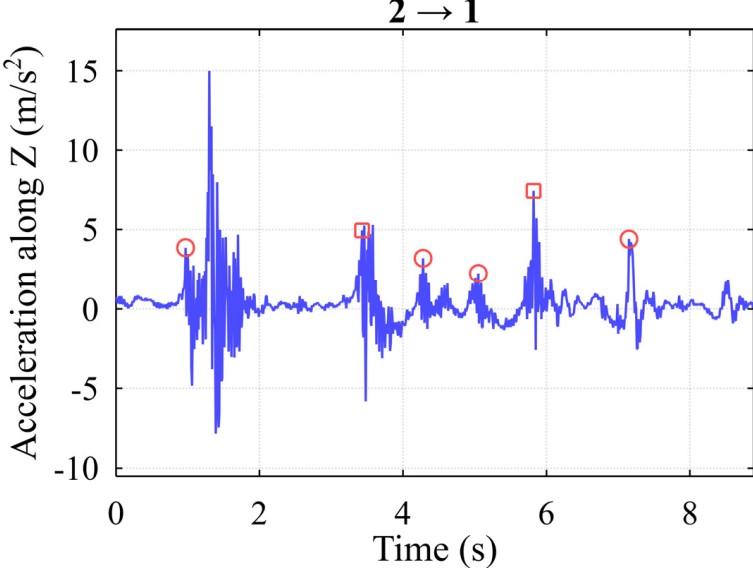

**Figure 25.** Primary and secondary impacts along the Z-axis (here the red circles indicate the initial impacts and the red squares indicate the secondary impacts).

The methodology confirmed the primary predictions: at the start of the 5th second, a second impact occurred due to spreader trajectory change and the start of the second hooking procedure. The following Figure 26 shows the histogram of potential impact events obtained after processing 102 cycles with the proposed methodology, indicating all the critical impacts during the aforementioned handling operations performed onboard the container ship (while moving the container along the vertical cell guides inside the ship towards the containers).

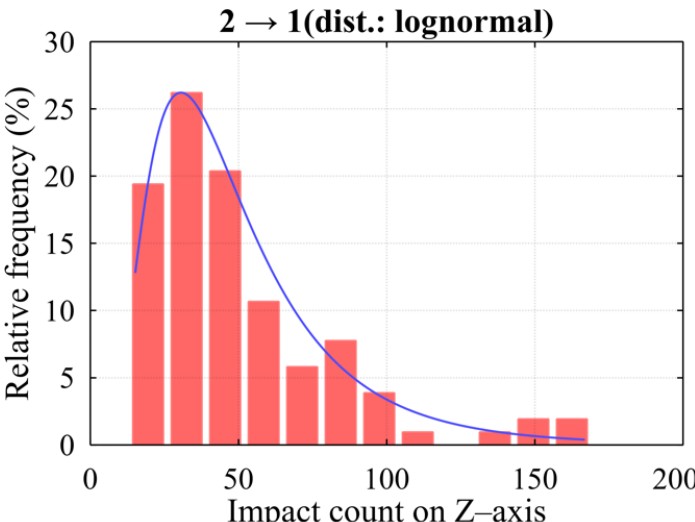

**Figure 26.** Histogram of potential impacts (events).

Results indicate that the number of overall impacts during these procedures is quite large, reaching up to 30 impact-induced vibrations in most cases (26.6%). The following histogram of recorded impacts according to the Z-axis (Figure 27) indicates that at least one impact event occurs at each new handling operation when the spreader impacts the container in the inner shaft of the ship, meaning that the methodology detects the initial hooking procedures well.

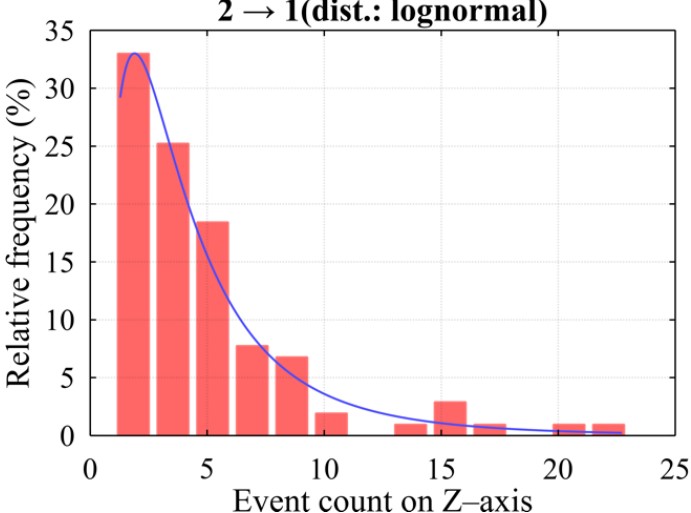

**Figure 27.** Histogram of detected impacts (events).

Almost 75% of all handling procedures are performed in blind conditions, reaching up to five impacts at a single procedure, which in turn results in metal deformations of different sorts, or at least strongly influences the construction integrity and safety in future operations. It can be argued that repeated impacts are not a rare phenomenon, and they occur quite regularly. With these precise experimental measurements, the algorithm detected enough impacts to the same areas of each corner to address the necessity of in-time detection and long-term Structural Health Monitoring. All of the impacts were detected with a high degree of accuracy using the presented step-by-step detection methodology and sensory technical means. The proposed methodology is therefore suitable to be used to detect these impacts in an almost real-time manner if used appropriately and with sufficient computational power for the inner logic to manage the analytics in due time. The following Figure 28 summarizes all the detected impacts and highlights the repeated impact during

the spreader lowering process toward the container in the inner shafts of the container ship.

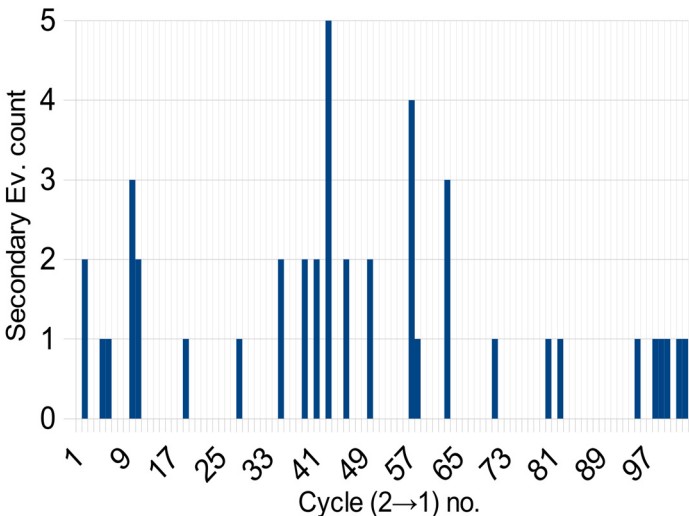

**Figure 28.** Comparison of the total number of detected impacts among the entire data set.

The results show that the proposed methodology can identify critical physical impacts, and in our case studies, we managed to detect up to five impacts at a single handling cycle, meaning that the spreader continuously impacted the same metal plate of the corner of the container near the hooking area, gradually deforming the metal structure due to continuous momentum stress. In other case studies, we managed to detect a higher number of impacts in almost half of the case studies, ranging between one and four impacts in most cases, indicating that almost 50% of the handling procedures are less efficient in terms of collision avoidance, due to the unpredictable human error factor and chaotic dynamics of the process due to external environmental influences. The results also show that the proposed solution can identify critical impacts in each case study.

**Author Contributions:** Conceptualization, S.J. and T.E.; methodology, S.J., T.E. and M.V.; software, P.P. and J.T.; validation, M.J. and P.P.; formal analysis, S.J. and T.E.; investigation, M.J. and M.V.; resources, M.J.; data curation, T.E.; writing—original draft preparation, S.J. and P.P.; writing—review and editing, M.V. and J.T.; visualization, T.E., P.P. and J.T.; supervision, S.J.; project administration, S.J.; and funding acquisition, S.J. All authors have read and agreed to the published version of the manuscript.

**Funding:** This research was funded by the Klaipeda University project JKSMART. This project has received funding from the European Regional Development Fund (project No. 01.2.2-LMT-K-718-03-0001) under a grant agreement with the Research Council of Lithuania (LMTLT).

**Institutional Review Board Statement:** Not applicable.

**Informed Consent Statement:** Not applicable.

**Data Availability Statement:** Not applicable.

**Conflicts of Interest:** The authors declare no conflict of interest.

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
