# Peer review of "Detecting Physical Impacts to the Corners of Shipping Containers during Handling Operations Performed by Quay Cranes"

_jmse, doi:10.3390/jmse11040794_

Round 1

Reviewer 1 Report

1.     The introduction section was recommended to be re-organized to better illustrated cutting-edge studies about the research problem. It seemed that the introduction was simply copied from a project report. Moreover, why there was a figure 7.4 in the introduction section.

2.     The authors were recommended to carefully proofread the manuscript due that many figures were numbered in a quite confusing manner, such as figure 96.

3.     Besides, how did the method identify abnormal event for the study?

4.     For figure 14.9, it was not clear how the peaks of acceleration oscillation obtained, and more explanations about the figure were in need.  

5.     The following studies were recommended to be properly cited: [1] Emerging marine pollution from container ship accidents: Risk characteristics, response strategies, and regulation advancements, Journal of Cleaner Production, Volume 376, 2022,134266. [2]Quantifying Arctic oil spilling event risk by integrating an analytic network process and a fuzzy comprehensive evaluation model, Ocean & Coastal Management, vol. 228, p. 106326, 2022/09/01/ 2022.

Author Response

Responses to reviewer 1:

  1. The introduction section was recommended to be reorganized to better illustrate cutting-edge studies about the research problem. It seemed that the introduction was simply copied from a project report. Moreover, why was there a figure 7.4 in the introduction section.
  • We thank the reviewer for this suggestion. Yet we think it is crucial for the reader to understand the nature of the problem, and all the main dynamic elements and get a short insight into the more recent advancement in signal processing, engineering, and general computer science. We have used use-case figures in this section to illustrate the actual dynamics so that the reader would be acquainted with the subject and the specifics.
  • We used literature on similar problems, and techniques, that are more or less in line with our topic. We have checked the paper and we did not detect figure 7.4. All numbering was done correctly.
  1. The authors were recommended to carefully proofread the manuscript due that many figures were numbered in a quite confusing manner, such as figure 96.
  • We have checked the paper and we did not detect these mistakes. Figures numbering was done correctly in the first submission. Perhaps this is influenced by the review system or the program being used. If the numbering is incorrect, it will be corrected in the final submission by the MDPI.
  1. Besides, how did the method identify the abnormal events for the study?
  • the acceleration signal is filtered by rejecting low frequencies (the "corner" frequency of the filter is selected using that analysis with 3D graphs);
    • all oscillation amplitudes are found in the filtered signal;
    • only those amplitudes (half-waves) that exceed the "threshold" are selected
    • these are the consequences of an "abnormal event" ("threshold" is selected from that analysis with 3D graphs);
    • the selected amplitudes are grouped, and those groups are considered a specific "abnormal event".
  1. For figure 14.9, it was not clear how the peaks of acceleration oscillation were obtained, and more explanations about the figure were in need.
  • We have made improvements clarifying figure 14 in the paper. Also, in the filtered signal we get oscillations, half-waves are distinguished in the oscillations using the "zero-crossing" method, and the maximum value of each half-wave is found -> "peaks of oscillation".
  1. The following studies were recommended to be properly cited: [1] Emerging marine pollution from container ship accidents: Risk characteristics, response strategies, and regulation advancements, Journal of Cleaner Production, Volume 376, 2022,134266. [2] Quantifying Arctic oil spilling event risk by integrating an analytic network process and a fuzzy comprehensive evaluation model, Ocean & Coastal Management, vol. 228, p. 106326, 2022/09/01/ 2022.
  • We have analyzed the proposed publications and we see that they can provide some more insight into the topic. We have added these two publications to our list.

Reviewer 2 Report

The study highlights the lack of attention to risks associated with handling shipping containers and increasing damages to their metal infrastructures. Furthermore, it aims to develop technical solutions and critical impact detection methods for handling containers by quay cranes. The study uses the Impacts Detection Methodology (IDM) to detect repeated impacts by the spreader hooking rods to container corners during hooking procedures. Moreover, actual statistics of these impacts in Klaipeda Port are analyzed and found that more than half of handling procedures carried a higher risk of structural damage. The study concludes that the IDM can be useful in detecting these impacts and that technical solutions are needed to reduce the risks associated with container handling.

However, the study looks like a technical paper rather than research paper. The Abstract should be as; research question, hypothesis, methodology, also with quantitative results Moreover the novelty of the study should be clear could be added to the last paragraph of the Introduction. The results and discussion are not presented well. In this context, results section should be revised with quantitative, discussed and compared with the results published in the literature, and speculations should be avoided. In addition to the main findings, the Conclusions section should indicate research gaps and research directions identified as the results of research presented. Moreover;

1- What is the contribution of the engineering literature? It should have explained explicitly.

2- Uncertainty of the results needs to be discussed, especially in the context of the main findings if needed.

3- How can the authors decide the cases? Have you sought expert opinion? If applied, what are the qualifications and quantities of the experts?

4- “Results indicate that more than half of handling procedures are carried out with a higher risk of structural damage to the container in the crucial area……”. This study is for Klapedia Port. Does it also apply to other container ports or is it just a technical report for Klapedia port?

Author Response

Responses to reviewer 2:

  1. The Abstract should be as; research question, hypothesis, methodology, also with quantitative results Moreover the novelty of the study should be clear and could be added to the last paragraph of the Introduction.
  • The abstract was corrected accordingly.
  1. The results and discussion are not presented well. In this context, the results section should be revised with quantitative, discussed, and compared with the results published in the literature, and speculations should be avoided.
  • We thank the reviewer for this suggestion. Yet we think that the results section already presents quantitative estimations based on use cases, with the most critical results presented. Recent literature does not provide any comparative statistics related to impact detection in containers in their corners during hooking procedures. This is a new area for research in transport engineering.
  1. In addition to the main findings, the Conclusions section should indicate research gaps and research directions identified as the results of the research presented.
  • While the Conclusions section is not required by the journal we have made specific structural improvements to the paper and eliminated material that is speculative and not necessary.

Moreover;

  1. What is the contribution of the engineering literature? It should have been explained explicitly.
  • Our problem has its own specifics. As practice shows, it is quite new to the engineering community from the problem area point of view, where the provided engineering literature suggests better research directions, advancements, and best practices in those fields.
  1. Uncertainty about the results needs to be discussed, especially in the context of the main findings if needed.
  • We are now working to acquire more statistics and our intent is to address these uncertainties in our future research from a more qualitative perspective.
  1. How can the authors decide the cases? Have you sought expert opinion? If applied, what are the qualifications and quantities of the experts?
  • This paper provides insights into the problem and is not focused on expert evaluation. We show that this problem exists, and it can be identified using provided methods and system components. Expert assessment will be done in further steps, accumulating responses from terminal personnel working directly with the transportation processes.
  1. “Results indicate that more than half of handling procedures are carried out with a higher risk of structural damage to the container in the crucial area……”. This study is for Klaipeda Port. Does it also apply to other container ports or is it just a technical report for Klaipeda port?
  • We have analyzed the case studies at Klaipeda Port. During communication with the authorities and analytics of the systems in use by other Ports in Europe, we can presume that similar things happen in other Ports as well. This paper is not a technical report, yet we tried to describe the problem area more precisely, with all the details, so that the reader not acquainted with these technologies could, in general terms, understand the topic and the solution.

Round 2

Reviewer 1 Report

My comments have been addressed.

Reviewer 2 Report

The authors could not answer the questions adequately in the revised manuscript. Their contribution to the engineering literature is not specified as well. The method, results and discussions are academically weak for the research paper. It is not suitable to be published as it is.